# Methods to Stimulate Sporulation and Freeze-Drying Strategies for the Conservation of *Diplodia mutila*, *Diplodia seriata*, *Lasiodiplodia theobromae,* and *Neofusicoccum arbuti* Isolated from Apple Trees with Canker and Dieback Symptoms

**DOI:** 10.3390/jof11090640

**Published:** 2025-08-29

**Authors:** Adrián Valdez-Tenezaca, Mauricio E. Gutiérrez, Matías Guerra, Jean Franco Castro, Sergio A. Covarrubias, Gonzalo A. Díaz

**Affiliations:** 1Laboratorio de Patología Frutal, Departamento de Producción Agrícola, Facultad de Ciencias Agrarias, Universidad de Talca, Campus Talca, Av. Lircay s/n, Talca 3460000, Chile; mauriciogutierrez@utalca.cl; 2Instituto de Investigaciones Agropecuarias (INIA) Quilamapu, Av. Vicente Méndez 515, Chillán 3800062, Chile; matias.guerra@inia.cl (M.G.); jean.castro@inia.cl (J.F.C.); 3Academic Unit of Chemical Sciences, Campus Siglo XXI, University of Zacatecas, Carretera Zacatecas-Guadalajara km 6, La Escondida, Zacatecas 98160, Mexico; sergio.hernandez@uaz.edu.mx

**Keywords:** *Botryosphaeria* spp., dieback, sporulation stimulation, freeze-drying, conidium, preservation

## Abstract

Four *Botryosphaeria* spp. isolated from apple with dieback symptoms, *Diplodia mutila*, *Diplodia seriata*, *Neofusicoccum arbuti,* and *Lasiodiplodia theobromae*, were subjected to different conditions to induce sporulation, then freeze-dried and stored in glass vials and ampoules at a temperature of 4 ° C using two protective media (skimmed milk powder in water at 20% and a mixture of skimmed milk powder at 20% plus 5% inositol). Viability was assessed after storage periods of 1, 90, 180, and 365 days. Low-nutrient growth conditions on 2% water agar supplemented with pine needles, incubated under UV light (λ = 350 nm) and at 24 ° C, effectively stimulated sporulation of all four *Botryosphaeriaceae* species. The survival rate of the isolates was similar. Overall viability showed slight but significant differences depending on the type of protective medium and storage container used for the freeze-dried cultures (*p* < 0.001). Among the tested media, the highest viability was maintained in vacuum-sealed glass ampoules using either a medium containing 20% skimmed milk powder with 5% inositol or 20% skimmed milk powder alone.

## 1. Introduction

Several species of *Botryosphaeriaceae,* including *Botryosphaeria*, *Diplodia*, *Lasiodiplodia*, and *Neofusicoccum,* have been reported to cause *Botryosphaeria* canker and dieback in apple trees worldwide [1,2,3,4,5,6,7,8,9]. In Chile, *Diplodia mutila*, *D. seriata*, *Lasiodiplodia theobromae,* and *Neofusicoccum arbuti* are the most frequently reported species causing *Botryosphaeria* canker and dieback in apple orchards [10].

Light has been considered a key stimulus for conidial formation in fungi. Exposure of mycelia to ultraviolet (UV) light has been a widely used method [11,12,13,14]. Ultraviolet light irradiation has successfully induced sporulation in *Ascomycetes* species [15], *Basidiomycetes* [16], *Myxomycetes* [17], and *Zygomycetes* [18]. Wavelengths between 350 and 500 nm have been reported to be the most effective for promoting sporulation [12]. However, different isolates may exhibit different threshold doses under specific wavelengths, while excessive exposure can inhibit sporulation [19,20]. Although the mechanism by which light induces fungal conidiation is not yet fully understood. Dahlberg and Etten et al. [12] suggested that photoreceptors might influence the electron transport system, affecting pH, cellular ionic balance, and glucose and carbohydrate metabolism. Several photosensitive proteins have been identified during fungal conidiation [18,21,22,23]; for instance, White Collar-1 and Vivid proteins have been recognized as blue light photoreceptors that mediate light induction of rhythmic conidiation in *Neurospora crassa* [24,25,26]. Furthermore, transcription factors such as BLR-1 and BLR-2 are essential in the photoconidiation process in *Trichoderma atroviride* [27].

Fungi are notoriously difficult to maintain under good conditions, as they are inherently variable [28]. Methods of preserving fungal strains as research tools require that stored cultures remain viable over time, without undergoing morphological or physiological changes. Among the various methods used for long-term storage of fungal cultures, some have been evaluated as more effective than others depending on the species or genus [29]. When preserving fungi, several factors must be considered, including economic aspects. Common methods include preservation under mineral oil or paraffin [30], in sterile water [30,31], in grain or soil at room temperature [30], in silica gel [30,32], and freezing at −20 °C or −80 °C [33]. However, it should be noted that cryopreservation at –20 °C is not recommended whereas preservation in liquid nitrogen (−196 °C) is considered an optimal method for long-term storage of fungal cultures due to the stability of secondary metabolite production and minimal genetic variations. Although certain fungi have shown significant polymorphism after recovery [34,35], cryopreservation is particularly suitable for the storage of valuable stock cultures, such as proprietary isolates [36].

Freeze-drying, or lyophilization, is a technique widely used by culture collections since it offers several advantages for the preservation of fungal cultures. For instance, it enables long-term storage stability, allowing samples to remain viable for decades, typically between 20 and 40 years, without significant viability loss. Freeze-dried materials can often be stored at room temperature, although temperatures between 4 and 18 °C are more suitable. The removal of moisture limits the risk of contamination and inhibits metabolic activity that could compromise the integrity of the preserved organisms, at the same time ensuring high viability and genetic stability upon rehydration. Another aspect to consider is that the resulting dried material is easy to handle and transport, which facilitates the exchange of cultures between laboratories and culture collections worldwide [36,37,38]. However, it has certain limitations, such as the time needed for recovery of the preserved isolate. To maintain optimal viability without sublethal damage, it requires the implementation of rigorous and controlled protocols, so the selection of an appropriate protective suspension medium, the optimal cooling rate, the cooling method, the temperature and speed of drying, and the final residual moisture content are critical to reduce damage to the fungal cell [28]. In this process, generally, only fungal conidia survive [28,35]. Skimmed milk is frequently used as a protective agent during freeze-drying, as it helps preserve the structural integrity of cell walls. Components of skimmed milk, such as amino acids, disaccharides, and calcium, create a protective layer around cells, which prevents damage caused by ice crystal formation during freezing and reduces stress during the drying phase. This protective effect is essential to maintain high cell viability, and the survival rate can be significantly improved compared with other protective agents [39,40,41]. Inositol is also used as a protective agent in mushroom freeze-drying due to its ability to stabilize cell walls during the freezing and drying process. This stabilization is crucial to preserve cell viability, as inositol acts to maintain the structural and functional integrity of cells by preventing ice crystal formation and other damage associated with the freeze-drying process [42]. Currently, freeze-drying remains one of the most effective methods for maintaining viable and stable fungal cultures for long periods. Numerous species of phytopathogenic fungi have been successfully preserved by freeze-drying [43,44,45]. It should be noted that no preservation method can be universally applied to all fungi and that intraspecific variability makes it impossible to apply standard protocols, even at the species level [28]. In particular, for *Botryosphaeria* spp., in addition to freeze-drying and cryopreservation in liquid nitrogen, the use of agar plug storage in mineral oil has been reported, which allowed the viability of unidentified *Botryosphaeria* spp. to be preserved for 31 years, *Botryosphaeria obtusa* for 20 years, and *Botryosphaeria ribis* for 6 years [30] (M. Fletcher, ICMP Landcare Research, Auckland, personal communication). Culture viability is a key indicator of success in fungal preservation methods. Although preservation of some *Botryosphaeriaceae* spp. has been described, the preservation by freeze-drying of species isolated from apple trees with dieback symptoms, such as *D. mutila*, *D. seriata*, *N. arbuti,* and *L. theobromae* has not yet been reported.

The objective of this study was to evaluate the sporulation stimulation conditions and the viability of the freeze-drying conservation method over time to preserve four *Botryosphaeria* spp. that affect apple trees, causing dieback and canker [10]. Maintaining the viability of fungal cultures is crucial to advancing research on the mechanisms of action, fungicide resistance, and pathogenicity of these phytopathogens.

## 2. Materials and Methods

### 2.1. Study Location

The experiments to stimulate conidial production were carried out at the Fruit Pathology Laboratory of the University of Talca, Chile, during 2023. The freeze-drying tests were carried out at the Microbial Genetic Resources Bank (BRGM), located at the Agricultural Research Institute (INIA) Quilamapu, Chillán, Chile, in the same year.

### 2.2. Isolation and Molecular Identification

In studies carried out by Díaz et al. [10], meticulous procedures for isolation, morphological characterization, molecular identification, and phylogenetic analysis of *D. mutila* (Bot-2017-DM21), *D. seriata* (Bot-2018-S3), *L. theobromae* (Bot-2018-LT45), and *N. arbuti* (Bot-2018-NA32) were performed. The isolation began with the collection of samples of infected plant tissue (Figure 1), which were subjected to a surface disinfection process and subsequent cultivation in selective media to obtain pure cultures. A total of 255 Botryosphaeriaceae-like isolates were obtained from 238 symptomatic positive samples [10]. The isolates obtained were morphologically characterized by observing the macroscopic characteristics of the colonies and the micromorphology of the conidia (Figure 2). For molecular identification, genomic DNA was extracted from the isolates using standardized commercial extraction protocols, and specific gene regions were amplified through polymerase chain reaction (PCR) [10]. In particular, molecular markers such as the internal transcribed spacer (ITS-1/4) region, translation elongation factor 1-α (tef1), and β-tubulin (tub2) were used [46,47,48] (Appendix A). The amplified products were sequenced, and the obtained sequences were compared by BLAST (v. 2.13.0) analysis against public databases, such as GenBank, for identification. Finally, phylogenetic analyses were performed to determine the evolutionary relationships between the isolates and other known members of the genus (Appendix A). These analyses combined molecular and morphological data, providing a robust and detailed identification of *D. mutila* (Bot-2017-DM21), *D. seriata* (Bot-2018-S3), *L. theobromae* (Bot-2018-LT45), and *N. arbuti* (Bot-2018-NA32). After molecular analysis, the following were identified: *D. mutila* (*n* = 6 isolates), *D. seriata* (*n* = 13 isolates), *L. theobromae* (*n* = 6 isolates), and *N. arbuti* (*n* = 8 isolates) [10]. It was then maintained in Acidified Potato Dextrose Agar (APDA) medium and stored in refrigeration at 4 °C until its use in the fungal collection of the Fruit Growing Laboratory of the University of Talca. Each isolate was recovered by seeding a mycelial plug (5 mm diameter) in 0.1% PDA medium plus 0.01% tetracycline hydrochloride, incubated at 25 ± 2 °C for 10 days. The isolates of *D. mutila* (Bot-2017-DM21), *D. seriata* (Bot-2018-S3), *L. theobromae* (Bot-2018-LT45), and *N. arbuti* (Bot-2018-NA32) were deposited as live cultures at the Chilean Collection of Microbial Genetic Resources in Chillán, Chile (www.cchrgm.cl, accessed on 10 August 2025), under the codes RGM 3110, RGM 3114, RGM 3136, and RGM 3138, respectively.

### 2.3. Conidial Production

Different growth conditions and media were evaluated to stimulate sporulation and conidial production in fungi of the *Botryosphaeriaceae* family, such as *D. mutila* (Bot-2017-DM21), *D. seriata* (Bot-2018-S3), *N. arbuti* (Bot-2018-NA32), and *L. theobromae* (Bot-2018-LT45), isolated from apple wood. Multiple growth conditions were considered, including exposure to temperature ranges between 5 and 24 °C. The type of growth medium and exposure to ultraviolet (UV) light at λ = 350 nm were evaluated. The media evaluated included potato dextrose agar (PDA), 2% agar–water (A-A), PDA with the addition of pine needles, PDA with the addition of apple chips, PDA with the addition of grapevine chips, A-A with the addition of pine needles, A-A with the addition of apple chips, A-A with the addition of grapevine chips, A-A in slant, and PDA in a 350 mL flask (the plant material was previously subjected separately to a 120 °C autoclave cycle for 20 min and subsequently added to each medium evaluated) (Appendix A). Ultraviolet light exposure was performed in a dark chamber using a UV lamp at λ = 350 nm for periods of 15 continuous days. Subsequently, the mature pycnidia from each treatment under a flow chamber were collected and crushed using sterile ceramic mortars in 20 mL of sterile water. The solution with conidia was then placed in 20 mL Falcon tubes and subjected to a centrifugation cycle of 5 min at 6 °C at 6000 rpm. The number of conidia was quantified in a Neubauer chamber.

### 2.4. Spore Collection, Protective Suspension Media, and Freeze-Drying

After the stimulation and conidia production process, petri dishes containing 2% water agar culture medium and pine needles were selected, as they showed greater conidia production in stimulation tests. A total of 20 petri dishes with 10, 60 mm pine needles per dish were used for *Diplodia mutila*, *Diplodia seriata*, *Neofusicoccum arbuti,* and *Lasiodiplodia theobromae*. Mature pycnidia were collected under a laminar flow hood and ground in 50 mL of sterile water using sterile ceramic mortars. The solution with conidia was then sieved with a 100-micron filter and then subjected to a centrifugation cycle of 10 min at 6 °C at 3000 rpm. The supernatant was removed, and the conidia concentrate was then resuspended with two protective solutions. The first consisted of skimmed milk powder in water at 20%, and the second solution was prepared with a mixture of skimmed milk powder at 20% plus 5% inositol. Both solutions were subsequently subjected to an autoclave cycle at 120 °C for 5 min. Approximately 0.5 mL of each suspension was added to two types of sterile containers: 2 mL glass vials and glass ampoules. The vial and glass ampoules were placed in a first cooling phase at a temperature of −20 °C for two hours. They were then placed on the racks of a primary lyophilizer Biobase BK-FD12PT (BIOBASE, Jinan, Shandong, China) under operating conditions of −60 °C and a vacuum pressure of 0.04 bar. The suspensions were dried for 15 h which left a residual moisture content of the dried suspension at 1–2%. The chamber was then brought to atmospheric pressure, and the vials and ampoules were removed. The group of containers, which were made up of glass vials, was then vacuum-sealed immediately after the freeze-drying process. However, the ampoules were fitted with a cotton plug, followed by a silica bead suspended above the sterilized cotton plug. The glass ampoules were then contracted approximately 1 cm above the top of the cotton plug and silica bead using a hydrogen-oxygen torch. This stage was carried out as quickly as possible to minimize the material being exposed to atmospheric oxygen and water vapor from the atmosphere, and these factors must be kept to a minimum; overexposure can cause deterioration. The glass ampoules were then placed in the manifold of a freeze-dryer CRIST 1-4 LSC basic (Martin Christ Gefriertrocknungsanlagen GmbH, Osterode am Harz, Germany) for a secondary drying phase at working conditions of 0.04 bar and −55 °C for 8 h. The ampoules were vacuum sealed using an air/gas torch [36]. Both vials and ampoules were stored at 4 °C.

### 2.5. Spore Viability

The viability of vials and ampoules containing freeze-dried *Botryosphaeriaceae* spp. was assessed after 1, 30, 180, and 365 days of preservation. The samples were placed at room temperature for two hours to assess their viability. Under a flow chamber, both types of containers were uncovered and reconstituted with 1000 µL of a 20% skimmed milk powder solution in each container, allowing 15–20 min for the spores to absorb moisture. The viability rate of freeze-dried *Botryosphaeriaceae* spp. was quantified by serial dilutions (10^−1^ to 10^−7^) in PDA. The strains were incubated at 24 °C for 24 h. The colony-forming units (CFU) produced by viable conidia were counted using a Nikon SMZ1500 Electronic Stereoscope (Nikon Corporation, Tokyo, Japan). The percentage reduction in viability was calculated using the formula $\%RV=\left.\left(1-\;\frac{CFUt}{CFU0}\right.\right)\times \;100$ CFU*t* = CFU count at storage time t (days). CFU*0* = initial colony-forming unit (CFU) count on day 1 (control). The percentage viability was analyzed using multifactorial ANOVA (*p* = 0.01) to identify significant effects of temperature, culture medium, supplements, UV, and their interactions. When overall differences were detected (*p* < 0.0001), Scheffé’s post hoc test (*p* < 0.0001) was applied for multiple comparisons between all experimental groups (including combinations of conditions).

### 2.6. Data Availability

The complete nucleotide sequence sets of DNA sequence for the ITS, tef1-α, and tub2 regions has been deposited in GenBank under accession numbers *Diplodia mutila* (MW560102, MW591887, MW574060), *Diplodia Seriata* (MH675471, MH908100, MH745086), *Lasiodiplodia Theobromae* (MW560112, MW591897, MW574070), and *Neofusicoccum arbuti* (MW560114, MW591899, MW574072) [10].

## 3. Results

### 3.1. Conidia Production

ANOVA analysis of variance showed that sporulation was significantly higher (*p*-value < 0.0001) in the four *Botryosphaeriaceae* spp. in 2% agar–water medium (W–A) in Petri dishes (PP), with the addition of apple chips at a growth temperature of 24 °C and with a 15-day exposure to ultraviolet light at λ = 350 nm, followed by (W–A) in Petri dishes (PD) with the addition of grapevine chips under the same temperature and “ultraviolet light conditions” as the most significant treatment. (W–A) in petri dishes (PD) with the addition of pine needles did not show significant differences with the second-best treatment and under the same temperature and ultraviolet light conditions (Figure 3) (Appendix A).

### 3.2. Lyophilization and Spore Viability

The freeze-drying process was evaluated using four *Botryosphaeriaceae* spp.: *D. mutila*, *D. seriata*, *L. theobromae,* and *N. arbuti*. The conidia of the described species were placed in two types of protective media before their freeze-drying process: the first, composed of a 20% skimmed milk powder solution in distilled water, and the second, consisting of 20% skimmed milk powder in distilled water supplemented with 5% inositol. The survival and viability results were evaluated after 1, 30, 180, and 365 days after the freeze-drying process. Using the two types of protective media, it was observed that the four *Botryosphaeriaceae* spp. evaluated and survived successfully. However, significant differences in long-term viability were observed between the two types of protective media used after an ANOVA analysis of variance (*p*-value < 0.0001) and a Scheffé statistical test (Appendix A). After 1 day of freeze-drying, the four *Botryosphaeriaceae* spp. freeze-dried using the two types of protective media described showed similar viability, indicating that both 20% skimmed milk powder alone and the combination with 5% inositol provided adequate initial protection. However, after 3 months, slight differences in the viability rate began to be observed. In the group treated with inositol, *N. arbuti* (Bot-2018-NA32) and *L. theobromae* (Bot-2018-LT45) showed greater stability and a smaller decrease in viability compared with the group that only used 20% skimmed milk. This trend was maintained and was accentuated at 180 and 365 days, where the viability of *N. arbuti* (Bot-2018-NA32) and *L. theobromae* (Bot-2018-LT45) in the group with inositol was slightly higher. *D. mutila* (Bot-2017-DM21) and *D. seriata* (Bot-2018-S3) showed similar viability values using the two types of protective media after 1, 30, 180, and 365 days after the freeze-drying process. In addition to the evaluation of the protective medium, two storage methods were compared: glass vials and glass ampoules. It was observed that the viability of *Botryosphaeriaceae* spp. was slightly lower in glass vials compared with glass ampoules, indicating that the fully airtight and vacuum-sealing of storage container also influenced the long-term survival of freeze-dried *Botryosphaeriaceae* spp. (Figure 4) (Appendix A).

## 4. Discussion

Fungi of the genus *Botryosphaeria* are important pathogens that affect a wide variety of fruit trees, causing diseases such as branch dieback and canker on trunks and main branches. These diseases are particularly devastating in high-value fruit trees, such as apple, pear, citrus, and grapevine, among others, generating significant losses in global fruit production [3,10]. The preservation of plant pathogenic fungi by freeze-drying is essential for scientific research and disease management. This method has allowed fungi to be stored in an inactive state, maintaining their viability and genetic characteristics in the long term without the need for continuous refrigeration, which, compared with the cryopreservation method, is less expensive and carries less risk of loss due to storage failures [36]. Freeze-drying can ensure the genetic purity of strains, which is essential for epidemiology studies, pathogenicity testing, and fungicide resistance evaluation. In addition, it facilitates the exchange of strains between institutions and laboratories, ensuring the availability of biological material for global research and the development of new disease control strategies. Considering the effectiveness of the freeze-drying method in the long-term preservation of plant-sporulating pathogenic fungi, the objective of this work was to establish the appropriate conditions to stimulate conidia production and evaluate the viability and survival for their conservation by the freeze-drying method in the species *D. mutila* (Bot-2017-DM21), *D. seriata* (Bot-2018-S3), *N. arbuti* (Bot-2018-NA32), and *L. theobromae* (Bot-2018-LT45) isolated from apple trees with dieback and canker symptoms in the Maule region, Chile.

The lack of sporulation of many fungi in culture is a problem frequently faced by mycologists and plant pathologists. This is particularly true in fungal identification, where the difficulty of identifying non-sporulating colonies is a major obstacle to developing adequate routine detection procedures or, failing that, for their use in laboratory and field experiments and especially for their preservation by cryopreservation and freeze-drying. Our study revealed that exposure to ultraviolet light (λ = 350 nm) of cultures of *Botryosphaeriaceae* spp. in 2% water agar alone was not sufficient to induce sporulation of the fungal species tested. Similarly, the addition of wood (pine needles, apple, and grape wood chips) to the culture medium without the presence of ultraviolet light did not produce significant sporulation. However, when both treatments were applied together (ultraviolet light (λ = 350 nm) and the addition of wood), robust sporulation was observed in *D. mutila* (Bot-2017-DM21), *D. seriata* (Bot-2018-S3), *N. arbuti* (Bot-2018-NA32), and *L. theobromae* (Bot-2018-LT45). This suggests that the lignin contained in wood could be acting as an inducer or sensitizer that, when interacting with ultraviolet radiation, activates certain metabolic or signalling pathways that are essential for cell differentiation and spore production [49,50]. Lignin is a complex and abundant polymer in nature, mainly in the cell walls of vascular plants such as apples, and its presence in the culture medium could be simulating natural environmental conditions that favor sporulation. In addition, ultraviolet radiation is known for its ability to generate oxidative stress, which can induce defense responses in fungi, including the production of reproductive structures such as conidia. The combination of both factors could be triggering an adaptive response in these fungal species, promoting sporulation as a survival strategy under adverse conditions [51,52,53]. Exposure to low temperatures did not influence sporulation or conidia production in *D. mutila* (Bot-2017-DM21), *D. seriata* (Bot-2018-S3), *N. arbuti* (Bot-2018-NA32), and *L. theobromae* (Bot-2018-LT45). This indicates that, unlike other fungi whose sporulation rates can be highly influenced by low temperatures [12,53,54], the *Botryosphaeriaceae* spp. evaluated in this study do not require a specific temperature to complete their sporulation cycle, as long as the other two key factors (UV light and lignin) are present. This observation may be related to the ability of these *Botryosphaeriaceae* spp. to adapt to diverse climates and environmental conditions, which reinforces their phytopathogenic character and their wide range of hosts in different geographical regions [55]. It should be noted that the sporulation of *Botryosphaeriaceae* spp. was higher when poor culture media, composed only of agar and water, were used compared with rich media such as PDA (Potato Dextrose Agar) medium. This observation suggests that a nutritious culture medium could be inhibiting or delaying the sporulation of these species. Under conditions of nutritional richness, fungi could prioritize vegetative growth over asexual reproduction, which would explain the lower production of conidia in media such as PDA. On the contrary, in a poor culture medium, the lack of essential nutrients could act as a stimulus that induces a stress response, promoting sporulation as a dispersal and survival strategy. Furthermore, our results highlight the importance of simulating minimal and non-enriched environmental conditions for sporulation studies, since fungi in their natural environment face nutritional limitation scenarios.

On the other hand, the use of 20% skimmed milk powder with the addition of 5% inositol as a protective medium in the freeze-drying process showed slight advantages in survival rate and a higher percentage of viability. The protective medium containing only 20% skimmed milk powder showed slightly lower viability and survival of conidia after a storage period of 365 days after the freeze-drying process. However, both protective media showed a similar survival rate for all *Botryosphaeriaceae* spp. This could be because inositol is known for its role in stabilizing cell membranes and preserving the integrity of cellular structures during the freeze-drying process [56]. In this study, it was shown that the incorporation of inositol into the protective medium with skimmed milk powder tended to improve the percentage of viability (RV) of conidia after lyophilization and prolonged storage compared with the exclusive use of skimmed milk powder. This observation could suggest that inositol could play a relevant role in cellular protection against the stress generated by the drying process and long-term storage conditions in *D. mutila* (Bot-2017-DM21), *D. seriata* (Bot-2018-S3), *N. arbuti* (Bot-2018-NA32) and *L. theobromae* (Bot-2018-LT45). Overall, it was observed that the protective medium was more effective when conjugated with the addition of 5% inositol, along with the use of glass ampoules as containers, significantly helping to reduce the viability loss. The addition of 5% inositol appeared to improve viability in all four *Botryosphaeriaceae* spp. This suggests that the addition of protective compounds, such as inositol, may contribute to a better preservation of the *Botryosphaeriaceae* spp. evaluated by reducing cell damage during dehydration and storage. This improvement was reflected in the higher number of CFU observed in the group treated with the combined medium and in the reduction of viability. Skimmed milk alone has been widely used as a protective medium due to its protein and carbohydrate content, which can help stabilize fungal cells during the freeze-drying process [28]. However, the results obtained in this study suggest that inositol increases viability when combined with skimmed milk, possibly due to its ability to interact with fungal cell lipids and proteins, minimizing damage caused by the freeze-drying process [57]. Skimmed milk is an ideal lioprotectant, which is inexpensive (standard skimmed milk, available at any grocery store, can also be used). The glass transition temperature of frozen skimmed milk is –18 °C, which facilitates freeze-drying, and it contains a mixture of macromolecules (lactalbumin and casein) and a saccharide. The macromolecules serve as filling agents; the amino acids contained in the macromolecules help repair sublethal damage and provide energy during rehydration. The saccharide lactose helps to decrease membrane transitions during dehydration by replacing water dipoles [58,59].

Regarding the type of container used, when comparing glass vials and ampoules as storage methods for the lyophilized product, slight significant differences were observed in the viability of the *Botryosphaeriaceae* spp. evaluated. Glass ampoules proved to be more effective in preserving viability after 365 days of storage. This finding may be related to the lower permeability of glass ampoules to moisture and air compared with glass vials, which contributed to better protection of the fungi against adverse environmental conditions. Exposure to moisture and temperature fluctuations can negatively affect the stability of lyophilized organisms [60]. Glass ampoules, by offering a more airtight seal, may have reduced these risks, resulting in greater preservation of the viability of the *Botryosphaeriaceae* spp. evaluated. The use of glass ampoules as a storage method proved to be a more robust container with better sealing properties and provided better protection during long-term storage. In contrast, glass vials presented a less efficient seal, allowing the entry of moisture and air, which may have compromised the viability of the lyophilized *Botryosphaeriaceae* spp. in this study.

In summary, this study concluded that the most effective conditions to induce sporulation in the evaluated *Botryosphaeriaceae* spp. were through their development in a poor agar–water culture medium with the addition of wood, under exposure to ultraviolet light at a temperature of 24 °C. It also demonstrated that during the freeze-drying process of *Botryosphaeriaceae* spp., the most effective protective medium was a skimmed milk solution supplemented with inositol, which was shown to maintain viability after long storage periods. This study is the first report describing specific methods to stimulate sporulation in *Botryosphaeriaceae* spp. and its subsequent preservation by freeze-drying process using isolates obtained from apple trees with dieback and canker symptoms in Chile.

## Figures and Tables

**Figure 1 jof-11-00640-f001:**
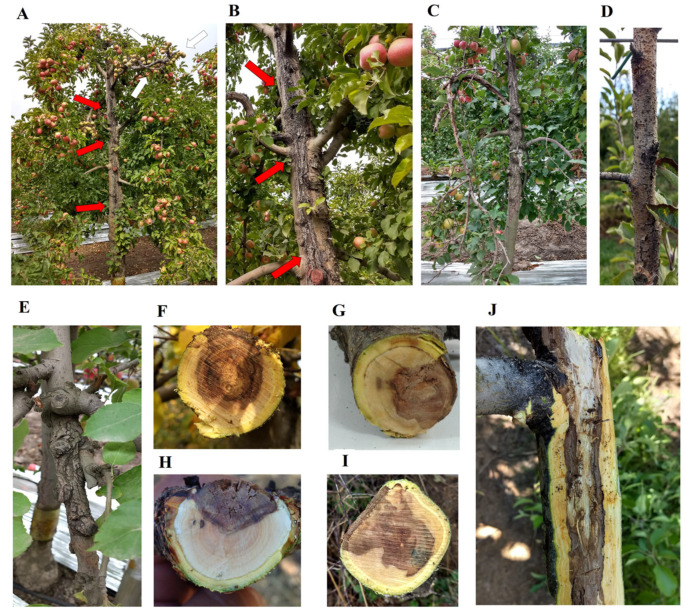
Symptoms of *Botryosphaeria* canker and dieback in commercial orchards of apple associated with *Botryosphaeriaceae* species isolated in central Chile. (**A**) Apple tree of 25-year-old cv. Cripps Pink with elongated canker in the trunk (red arrow) and dieback (white arrow). (**B**) Severe perennial cankers in the trunk of a 25-year-old apple tree cv. Cripps Pink. (**C**) Mature canker in the trunk and branches with dieback (15-year-old tree cv. Cripps Pink). (**D**) Young tree (7-year-old) cv. Fuji showing orange canker and dieback with presence of pycnidia and of central axis. (**E**) perennial canker in secondary branch of apple tree cv. Fuji. (**F**–**H**) Cross-section of branches with dieback showing brown hard cankers (U- and V-shaped). (**I**) Cross-section of trunk of mature tree of a 25-year-old cv. Gala showing brown hard V-shaped canker. (**J**) Longitudinal section of dieback branch of 12-year-old cv. Fuji showing elongated brown hard necrosis in the wood. *Diplodia mutila* (**F**), *Diplodia seriata* (**G**), *Lasiodiplodia theobromae* (**H**), and *Neofusicoccum arbuti* (**I**) show similar symptoms that are characteristic of *Botryosphaeria* dieback, generating wedge-shaped dark brown to blackish wood necrosis [10].

**Figure 2 jof-11-00640-f002:**
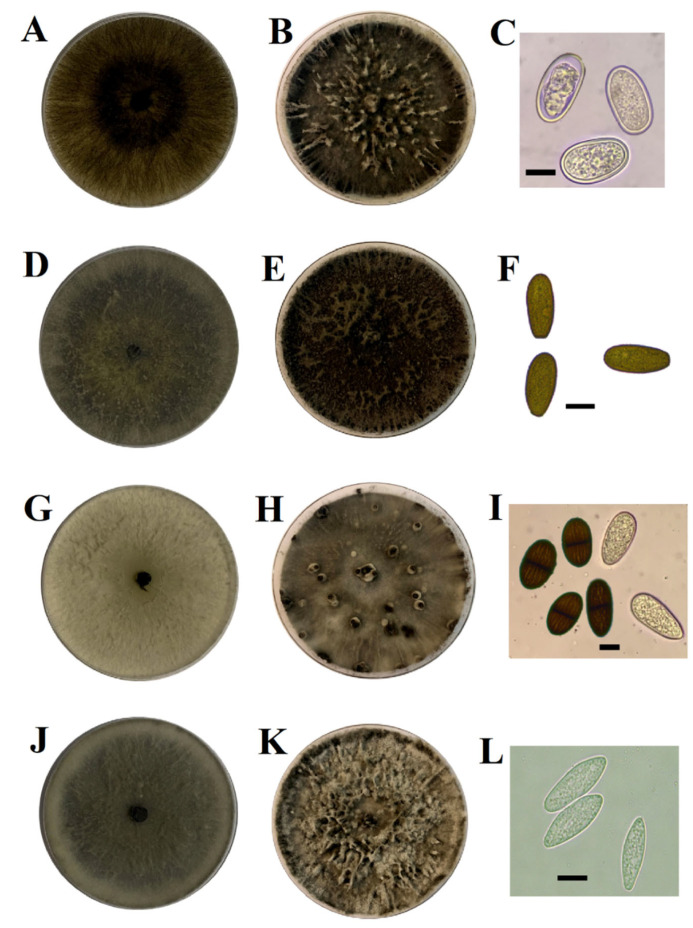
Colony and conidial morphology of *Botryosphaeriaceae* species obtained from *Botryosphaeria* canker and dieback on apple trees in central Chile. (**A**–**C**) *Diplodia mutila* isolate Bot-2017-DM21 (RGM 3110) showing colony of 7-day-old (**A**) and 21-day-old (**B**) on APDA at 25 °C, and hyaline, thick-walled, unicellular, and ellipsoidal to cylindrical conidia (**C**). (**D**–**F**) *D. seriata* isolate Bot-2018-S3 (RGM 3114) showing colonies of 7-day-old (**D**) and 21-days-old (**E**) on APDA at 25 °C, and mature brown, aseptate, and rounded conidia but with a truncate base (**F**). (**G**–**I**) *Lasiodiplodia theobromae* isolate Bot-2017-LT6 (RGM 3136), showing colony of 7-day-old (**G**) and 21-day-old (**H**) on APDA at 25 °C, and hyaline, aseptate, immature conidia along with mature dark brown condidia with longitudinal striations and central septum (**I**). (**J**–**L**) *Neofusicoccum arbuti* isolate Bot-2018-LT32 (RGM 3138) showing colony of 7-day-old (**J**) and 21-day-old (**K**) on APDA at 25 °C, and hyaline, aseptate, fusiform conidia with truncated base (**L**). Scale bar = 10 µm [10].

**Figure 3 jof-11-00640-f003:**
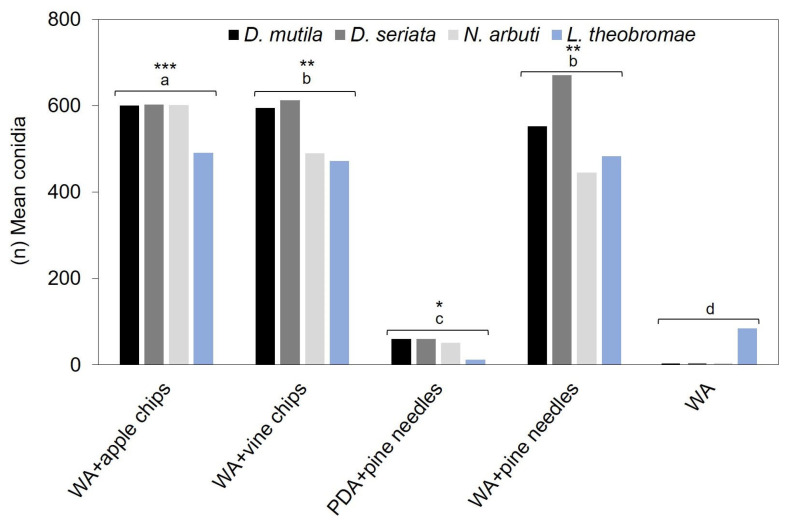
Average conidia production in *Diplodia mutila*, *Diplodia seriata*, *Neofusicoccum arbuti*, and *Lasiodiplodia. theobromae* under different substrates and growth conditions. Bars show culture media based on water agar (WA) with the addition of apple chips, grape chips, pine needles y PDA medium with pine needles, and water agar (WA) without supplements. Development temperature was 24 °C and 15 days of exposure to ultraviolet light (λ = 350 nm) under a dark chamber. (*) Shows the relative level of conidia production, where * represents low, ** intermediate and *** high production. Values with the same letters are not significantly different according to the Tukey test (*p* value < 0.0001) when comparing different culture substrates.

**Figure 4 jof-11-00640-f004:**
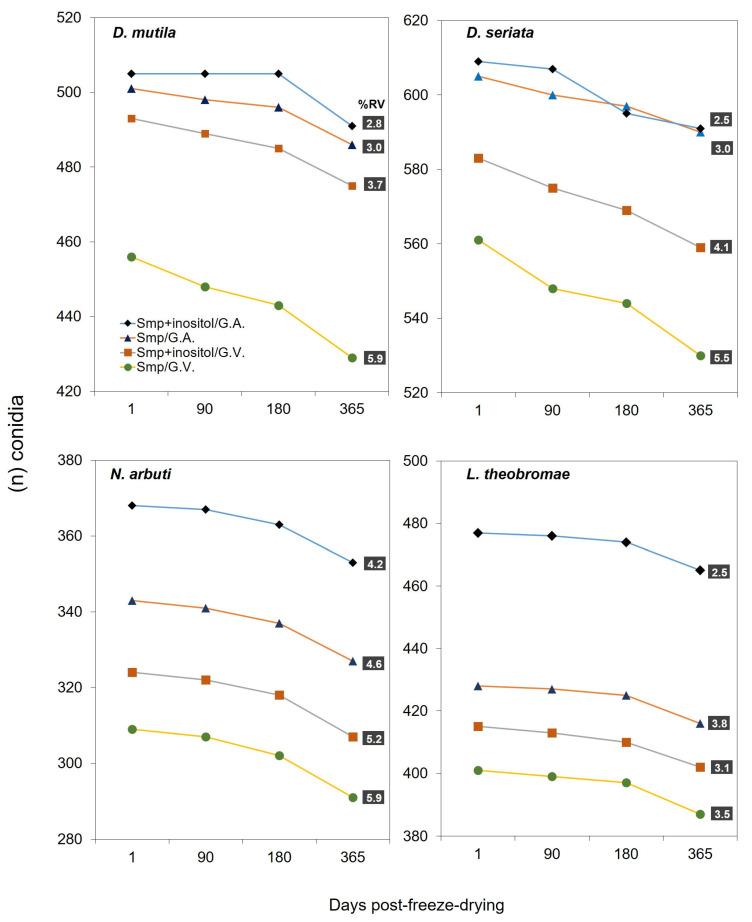
Viability and survival of *Diplodia mutila*, *Diplodia seriata*, *Neofusicoccum arbuti,* and *Lasiodiplodia theobromae* after freeze-drying. The number of colony-forming units (CFU) counted at a 10^−4^ post-freeze-drying dilution is shown. Type of container used for the freeze-drying process: G.V. = glass vials and G.A. = glass ampoules. Protective medium used for the freeze-drying process: (SMP) 20% skimmed milk powder. (SMP + inositol) 20% evaporated skimmed milk plus the addition of 5% inositol. (RV) represents the percentage reduction in viability after the freeze-drying process, over time (after 365 days of storage).

## Data Availability

The complete nucleotide sequence sets of DNA sequence for the *ITS*, *tef1-α*, and *tub2* regions have been deposited in GenBank under accession numbers: *D. mutila* (MW560102, MW591887, MW574060), *D. Seriata* (MH675471, MH908100, MH745086), *L. Theobromae* (MW560112, MW591897, MW574070), and *N. arbuti* (MW560114, MW591899, MW574072).

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
