# Peer review of "Methods to Stimulate Sporulation and Freeze-Drying Strategies for the Conservation of Diplodia mutila, Diplodia seriata, Lasiodiplodia theobromae, and Neofusicoccum arbuti Isolated from Apple Trees with Canker and Dieback Symptoms"

_jof, 2025, doi:10.3390/jof11090640_

Round 1
Reviewer 1 Report
The manuscript (jof_3810709) describe new methods to stimulate sporulation and freeze-drying strategies for the conservation of Diplodia mutila, Diplodia seriata, Lasiodiplodia theobromae and Neofusicoccum arbuti isolated from apple trees with canker and dieback symptoms. These tecnics can help researcher that need abundant sporulation of Botryosphaeriaceae species or want require that stored cultures remain viable over time. Future investigations of others researchrd should be validate this methodology by use in yours studies, because this genus is hard to sporulation in culture median.
Although, I had found that this manuscript might be acceptable for publication in Journal of Fungi the present text can be improved in some points that are detached within of the archive pdf. I highlight some points below, such as: Change the title; Some important parts of the methodology were omitted; In results some statistical data should be revised. Some figures (figures 2 and 4) should be change to supplementary, and some supplementary (supplementary figure 1) should be transformed in figure in the text; Legend of figures and tables should be self-explanatory. Currently the legends of all tables and figures are insufficient. Result’s pieces were added in the discussion, including the cittation of figures, thus I suggest the autohors correct the mistakes. Thus, I recommend MAJOR revision to this manuscrit.
Others comments/suggestions can be found as track changes in the other document.
Dear Dr (a) Rita Jiang
The manuscript (jof_3810709) describe new methods to stimulate sporulation and freeze-drying strategies for the conservation of Diplodia mutila, Diplodia seriata, Lasiodiplodia theobromae and Neofusicoccum arbuti isolated from apple trees with canker and dieback symptoms. These tecnics can help researcher that need abundant sporulation of Botryosphaeriaceae species or want require that stored cultures remain viable over time. Future investigations of others researchrd should be validate this methodology by use in yours studies, because this genus is hard to sporulation in culture median.
Although, I had found that this manuscript might be acceptable for publication in Journal of Fungi the present text can be improved in some points that are detached within of the archive pdf. I highlight some points below, such as: Change the title; Some important parts of the methodology were omitted; In results some statistical data should be revised. Some figures (figures 2 and 4) should be change to supplementary, and some supplementary (supplementary figure 1) should be transformed in figure in the text; Legend of figures and tables should be self-explanatory. Currently the legends of all tables and figures are insufficient. Result’s pieces were added in the discussion, including the cittation of figures, thus I suggest the autohors correct the mistakes. Thus, I recommend MAJOR revision to this manuscrit.
Others comments/suggestions can be found as track changes in the other document.

Author Response
Dear Reviewer 1
We sincerely appreciate your comments and observations on our manuscript and the time
it took to review it. We have carefully reviewed each of your recommendations and made
the corresponding modifications. We believe that all the comments have been satisfactorily
incorporated.
Regarding the question about the possibility of performing statistical analyses to compare
treatments within time intervals, we would like to explain that the main objective of our
research was to evaluate the long-term preservation of conidia after the lyophilization
process. For this reason, demonstrating statistical significance within shorter time periods
might not adequately reflect the overall behavior of viability over a long period. Instead, we
have chosen to use the percentage reduction in viability (%RV) as an indicator, as it allows
a clear and direct estimate of conidia viability over a prudent storage period.
We are convinced that these modifications fully reflect your comments and contribute
significantly to the final quality of the article. We thank you again for your valuable time. The
suggestions made are highlighted in blue in the text.
Sincerely,
the authors

Reviewer 2 Report
The article by the respected authors is an interesting study aimed at creating conditions for stimulating the formation of conidia of the species Diplodia mutila, Diplodia seriata, Lasiodiplodia theobromae and Neofusicoccum arbuti, assessing their viability and survival for preservation by freeze-drying. In my opinion, this work is relevant and will be of interest to mycologists and phytopathologists. In mycological studies, the problem of preserving fungal strains, while maintaining their viability without morphological changes, is acute.
The work is original, the title and abstract correspond to the content of the article. I would like to note the very high-quality illustrations that complement the text well.
There are several comments on the article.
1. In the "Introduction" it is better to give a clearly formulated goal of the work at the end (it is given in the "Discussion" section - lines 348-352).
2. In "Materials and Methods" - it is unclear how many isolates were obtained. "The isolates obtained were morphologically characterized by observing the macroscopic characteristics of the
colonies and the micromorphology of the conidia" - lines 127-129. Question - how many isolates were obtained?
3. It is necessary to give the names of the primers used in the work and their sequence. "In particular, molecular markers such as the internal transcribed spacer (ITS) region, translation elongation factor 1-α (tef1), and β-tubulin (tub2) were used." - lines 132-133. Which ITS? ITS 1/4 or ITS1/2? Need to clarify. Provide a table - primer name, sequence, source.
4. It would be better to add a "Conclusion" at the end of the article. It is already written (lines 449 - 473), so it will not be difficult for the authors to format it.
Overall, I believe that the article can be accepted for publication after making some minor edits.
The article by the respected authors is an interesting study aimed at creating conditions for stimulating the formation of conidia of the species Diplodia mutila, Diplodia seriata, Lasiodiplodia theobromae and Neofusicoccum arbuti, assessing their viability and survival for preservation by freeze-drying. In my opinion, this work is relevant and will be of interest to mycologists and phytopathologists. In mycological studies, the problem of preserving fungal strains, while maintaining their viability without morphological changes, is acute.
The work is original, the title and abstract correspond to the content of the article. I would like to note the very high-quality illustrations that complement the text well.
There are several comments on the article.
1. In the "Introduction" it is better to give a clearly formulated goal of the work at the end (it is given in the "Discussion" section - lines 348-352).
2. In "Materials and Methods" - it is unclear how many isolates were obtained. "The isolates obtained were morphologically characterized by observing the macroscopic characteristics of the
colonies and the micromorphology of the conidia" - lines 127-129. Question - how many isolates were obtained?
3. It is necessary to give the names of the primers used in the work and their sequence. "In particular, molecular markers such as the internal transcribed spacer (ITS) region, translation elongation factor 1-α (tef1), and β-tubulin (tub2) were used." - lines 132-133. Which ITS? ITS 1/4 or ITS1/2? Need to clarify. Provide a table - primer name, sequence, source.
4. It would be better to add a "Conclusion" at the end of the article. It is already written (lines 449 - 473), so it will not be difficult for the authors to format it.
Overall, I believe that the article can be accepted for publication after making some minor edits.
Author Response
Dear Reviewer 2
We sincerely appreciate the time and attention you dedicated to reviewing our
manuscript. We have carefully considered each of your comments and made the
appropriate modifications to improve the clarity and robustness of the work.
1. As suggested, we have included the main objective in the "Introduction" section
at the end of that section, in order to present it clearly and distinctly.
2. In the "Materials and Methods" section, the exact number of isolates obtained and
characterized has been specified.
3. We have incorporated a specific table in the context of the manuscript detailing
the names of the primers used, their sequences (in 5'–3' orientation), and the
corresponding original reference, including clarification of the ITS region used
(ITS1/ITS4).
4. Finally, the "Conclusions" section has been formally incorporated at the end of the
manuscript.
We are convinced that these modifications fully reflect your comments and contribute
significantly to the final quality of the article. Suggestions are highlighted in the text
with blue color.
Sincerely,
The authors

Round 2
Reviewer 1 Report
Dear Dr (a) Rita Jiang
The manuscript (jof_3810709) describe new methods to stimulate sporulation and freeze-drying strategies for the conservation of Diplodia mutila, Diplodia seriata, Lasiodiplodia theobromae and Neofusicoccum arbuti isolated from apple trees with canker and dieback symptoms. These methodologies can help researcher that need abundant sporulation of Botryosphaeriaceae species or want require that stored cultures remain viable over time. Future investigations of others researchers should be validate this methodology by use in yours studies, because this genus is hard to sporulation in culture median.
I had found that this manuscript might be acceptable for publication in Journal of Fungi, however the present text can be improved in some points that are detached within of the archive pdf. I highlight some points below: In figure 1 associate the symptom showed with fungal specie isolated. The supplementary (supplementary S1) should be transformed in figure in the text; Revise a/or Remake the statistical analysis of the figure 2. Thus, I recommend MINOR revision to this manuscript.
Others comments/suggestions can be found as track changes in the attached document.
Dear Dr (a) Rita Jiang
The manuscript (jof_3810709) describe new methods to stimulate sporulation and freeze-drying strategies for the conservation of Diplodia mutila, Diplodia seriata, Lasiodiplodia theobromae and Neofusicoccum arbuti isolated from apple trees with canker and dieback symptoms. These methodologies can help researcher that need abundant sporulation of Botryosphaeriaceae species or want require that stored cultures remain viable over time. Future investigations of others researchers should be validate this methodology by use in yours studies, because this genus is hard to sporulation in culture median.
I had found that this manuscript might be acceptable for publication in Journal of Fungi, however the present text can be improved in some points that are detached within of the archive pdf. I highlight some points below: In figure 1 associate the symptom showed with fungal specie isolated. The supplementary (supplementary S1) should be transformed in figure in the text; Revise a/or Remake the statistical analysis of the figure 2. Thus, I recommend MINOR revision to this manuscript.
Others comments/suggestions can be found as track changes in the attached document.

Author Response
Dear Reviewer:
We deeply appreciate your valuable comments and the time you spent reviewing our manuscript.
We have carefully considered each of the points raised and made the corresponding modifications
to improve the clarity of the work.
ï‚· Regarding Figure 1, the explicit association between the observed symptom and the isolated
fungal species has been incorporated, as you suggested.
ï‚· Regarding the supplementary material (S1), it has been transformed into a figure integrated
into the body of the text, so that the reader can access the information in a more direct and
understandable way.
ï‚· Finally, we have revised and redrafted the statistical analysis in Figure 3, which shows the
results more clearly. We should note that the statistical results compare the conidia
production produced on each substrate by grouping the four Botryosphaeria species together
and do not compare conidia production by species, as highlighted in the main text.
ï‚· The changes are highlighted in blue
We trust that the implemented changes adequately address your recommendations and
significantly contribute to the quality of the manuscript. We are very grateful for your comments,
which have been of great help in strengthening our work.
Sincerely
The authors